# Towards Zero-Waste Valorization of African Catfish By-Products Through Integrated Biotechnological Processing and Life Cycle Assessment

**DOI:** 10.3390/gels12010045

**Published:** 2026-01-01

**Authors:** Orsolya Bystricky-Berezvai, Miroslava Kovářová, Daniel Kašík, Ondřej Rudolf, Robert Gál, Jana Pavlačková, Pavel Mokrejš

**Affiliations:** 1Department of Polymer Engineering, Faculty of Technology, Tomas Bata University in Zlín, Vavrečkova 5669, 760 01 Zlín, Czech Republic; bystricky_berezvai@utb.cz; 2Centre of Polymer Systems, Tomas Bata University in Zlín, Třída Tomáše Bati 5678, 760 01 Zlín, Czech Republic; kovarova@utb.cz; 3Department of Chemistry, Faculty of Technology, Tomas Bata University in Zlín, Vavrečkova 5669, 760 01 Zlín, Czech Republic; d_kasik@utb.cz; 4Department of Lipids, Detergents and Cosmetics Technology, Faculty of Technology, Tomas Bata University in Zlín, Vavrečkova 5669, 760 01 Zlín, Czech Republic; rudolf@utb.cz (O.R.); pavlackova@utb.cz (J.P.); 5Department of Food Technology, Faculty of Technology, Tomas Bata University in Zlín, Vavrečkova 5669, 760 01 Zlín, Czech Republic; gal@utb.cz

**Keywords:** multiproduct extraction, biotechnological treatment, by-products, gelatin, hydrolysate, oil, pigment, zero waste, life cycle assessment, sustainability

## Abstract

African catfish (*Clarias gariepinus*, AC) is one of the most widely farmed freshwater fish species in Central Europe. Processing operations generate up to 55% by-products (BPs), predominantly carcasses rich in proteins, lipids, and minerals. This study develops a comprehensive valorization process for ACBPs to recover gelatin, protein hydrolysate, fish oil, and pigments. The processing protocol consisted of sequential washing, oil extraction, demineralization, and biotechnological treatment to disrupt the collagen quaternary structure. A two-factor experimental design was employed to optimize the processing conditions. The factors included the extraction temperatures of the first (35–45 °C) and second fraction (50–60 °C). We hypothesized that enzymatic conditioning, combined with sequential hot-water extraction, would yield gelatin with properties comparable to those of mammalian- and fish-derived gelatins, while enabling a near-zero-waste process. The integrated process yielded 18.2 ± 1.2% fish oil, 9.8 ± 2.1% protein hydrolysate, 1.7 ± 0.7% pigment extract, and 25.3–37.8% gelatin. Optimal conditions (35 °C/60 °C) produced gelatin with gel strength of 168.8 ± 3.6 Bloom, dynamic viscosity of 2.48 ± 0.02 mPa·s, and yield of 34.76 ± 1.95%. Life cycle assessment (LCA) identified two primary environmental hotspots: water consumption and energy demand. This near-zero-waste biorefinery demonstrates the potential for comprehensive valorization of aquaculture BPs into multiple value-added bioproducts.

## 1. Introduction

The global fish farming industry is experiencing rapid growth, with production projected to reach 205 million tons by 2032, representing a 10% increase from 2022 [1]. This expansion results in the generation of substantial BPs, which can account for 30% to 70% of the wet weight of processed fish, depending on the species. Efficient valorization of these BPs is essential to improve the sustainability and profitability of fish processing operations [2].

AC, a warm-water fish species, has emerged as a species of significant economic interest due to its rapid growth at high stocking densities, omnivorous feeding habits, adaptability, disease resistance, and the tastiness of its flesh [3,4]. Global production of AC has increased considerably, with Nigeria being the largest producer, reporting an annual output of approximately 1 million metric tons valued at around USD 2.6 billion [5]. Hungary leads in AC production in Europe, yielding over 5100 tons per year [6]. The volume of BPs from AC varies depending on regional consumption habits and processing practices, reaching up to 55% of the total fish weight. Valorizing ACBPs is closely aligned with circular economy principles, which focus on reducing, reusing, recycling, and recovering materials to minimize waste and environmental impact. Implementing zero-waste technologies enables the extraction of valuable components from ACBPs, which are rich in protein (18.34–20.32%), polyunsaturated fatty acids (PUFAs), minerals, and pigments [7,8,9].

Enzymatic hydrolysis represents a sustainable method for converting collagen-rich waste into low-molecular-weight bioactive peptides [9]. Proteolytic enzymes, which hydrolyze peptide bonds, operate under mild conditions, reducing energy consumption and minimizing waste compared to traditional chemical methods (acid or alkali pretreatment of the raw material), which often produce high-salt BPs that complicate disposal [10,11,12]. Enzymes selectively cleave peptide bonds, preserving amino acids essential for gelatin’s gelling properties.

Proteases are primarily classified based on their source of origin. They may be derived from microorganisms, such as bacteria, fungi, and viruses (e.g., Protamex^®^, Alcalase^®^, Corolase^®^), or obtained from plants (e.g., papain, bromelain) and animals (e.g., trypsin, pepsin) [13]. Microbial proteases are particularly advantageous due to their rapid production, minimal space requirements, and the potential for genetic modification to enhance enzyme characteristics [12]. Proteases can also be categorized based on their site of action, with endopeptidases cleaving internal peptide bonds and exopeptidases acting at the ends of peptide chains [13].

Gelatin is a versatile, high-molecular-weight macromolecule composed of three polypeptide strands derived through partial hydrolysis of collagen, the main structural protein in white fibrous connective tissues. It is well known for its excellent gelling, thickening, foaming, and stabilizing properties, which arise from its coil-helix transition. Traditionally, gelatin is sourced from porcine or bovine origins [14,15,16]. Fish gelatin is increasingly recognized as an alternative to bovine and porcine gelatin, driven by religious, ethical, and health considerations [17,18]. Notable differences exist between gelatins derived from cold-water and warm-water fish. Cold-water fish gelatin typically has lower hydroxyproline content, resulting in lower gel modulus (low gel strength), denaturation temperature, melting and gelling temperatures (below 8–10 °C), and thermal shrinkage, making it suitable for microencapsulation applications [15,18,19]. Warm-water fish gelatin, such as that obtained from AC, has favorable functional properties, including good gel strength, emulsifying capacity, film-forming abilities, reasonable thermal stability, higher melting point (25–27 °C), and high bioactivity, making it suitable for food, pharmaceutical, and biomedical applications [14,17,20]. However, it still faces challenges such as a higher degradation rate, lower thermal stability, reduced gel strength, and lower gelling and melting temperatures compared to mammalian gelatin (32–35 °C). These limitations are primarily due to its lower content of proline and hydroxyproline—amino acids essential for a more substantial gelling effect—while fish gelatins exhibit greater variability in their amino acid composition [18,20]. As a result, products made from warm-water fish gelatin generally require storage at low temperatures [18]. Despite its versatility and functional properties, the large-scale development of fish gelatin is constrained by weaker rheological properties, inconsistent raw material quality, stringent legislation, and issues related to odor and color [21,22].

Beyond gelatin, other valuable BPs can be recovered from AC. Fish oil derived from AC is rich in essential fatty acids (FAs), including oleic acid (18:1), omega-3 PUFAs such as eicosapentaenoic acid (EPA; C20:5(n-3)), docosahexaenoic acid (DHA; C22:6(n-3)), and α-linolenic acid (ALA; 18:3(n−3)), as well as omega-6 FA like linoleic acid (LA, 18:2(n−6)). The most abundant saturated fatty acids (SFAs) in AC oil are palmitic acid (PA, 16:0) and stearic acid (SA, 18:0) [23,24,25]. Additionally, fish oil is a good source of fat-soluble vitamins such as A and D [21]. Due to its rich composition, fish oil is valuable for nutraceutical, food, cosmetic, and bio-based industrial applications. Its bioactive compounds support cardiovascular and cognitive health, regulating blood pressure and preventing type II diabetes, alleviating the severity of photoaging, skin cancer, allergies, dermatitis, cutaneous wounds, acne, and melanogenesis [18,23,26,27].

AC bones, primarily composed of calcium phosphate, can be used for calcium supplements, organic fertilizers, and functional foods [28]. Apart from calcium, magnesium, and sodium, trace elements as manganese, zinc, iron, copper, iodine, and cobalt can also be found in AC [9]. Pigments in AC skin, although underexplored, hold potential as natural colorants. The pigmentation, FA, and mineral composition can vary depending on diet, environment, and growth stage [9,29,30]. Furthermore, solid residues from gelatin extraction can be repurposed as animal feed or fish sauce, enhancing sustainability [22].

Food production is responsible for about one-fourth of global greenhouse gas emissions, highlighting the urgent need to reduce emissions, water usage, and waste generation [10,11]. Adopting circular economy models and data-driven management in the seafood sector can significantly enhance efficiency and sustainability. Utilizing seafood waste in other industries and reintegrating these materials into production cycles helps create sustainable production loops. LCA is a critical tool for evaluating and managing resource use, emissions, and environmental impacts throughout the product life cycle [10,31]. LCA is well established for crops and livestock, but its application to fish-based products such as gelatin remains limited, partly due to the lack of comprehensive inventory data for novel or region-specific materials. Existing studies show that gelatin production using acid or alkali conditioning of collagen can significantly contribute to eutrophication-causing emissions, yet detailed LCA data are insufficient [32]. Early-stage LCA can provide valuable guidance for eco-efficient fish-processing practices and supports progress toward the United Nations Sustainable Development Goal 14 (Life Below Water) [10]. Innovations such as enzymatic hydrolysis provide greener alternatives to traditional gelatin extraction methods, reducing waste, processing time, freshwater utilization, and chemical use [33].

The primary objective of this study is to address a notable gap in the literature by valorizing underexploited African catfish by-products (ACBPs) through an integrated process for recovering gelatin, protein hydrolysates, oil, and pigments, utilizing a novel biotechnological strategy based on the enzymatic conditioning of purified collagen with microbial proteases. This work also investigates how selected technological parameters influence the properties of the resulting gelatin fractions. Although African catfish is widely farmed in aquaculture and generates substantial processing waste, its by-products remain largely unexploited. This is particularly relevant in the context of Sustainable Development Goal 14, which aims to eliminate marine discards by 2030. Currently, most solid fish waste is disposed of through incineration, composting, anaerobic digestion, landfilling, or even sea dumping—practices that may pose environmental and health risks [34]. While a portion of fish waste is converted into low-value feed ingredients, only a limited amount is transformed into high-value bioproducts, and no comprehensive valorization strategy has been proposed for AC by-products. To address this gap and reinforce sustainability principles, a zero-waste processing concept was implemented and evaluated for the first time on ACBPs using laboratory-scale life cycle assessment (LCA). The assessment demonstrates alignment with circular bioeconomy objectives and identifies environmental hotspots for process improvement. The working hypothesis posits that applying a comprehensive processing technology—comprising collagen purification, enzymatic conditioning, and sequential hot-water extraction—will yield gelatin with extraction efficiencies and quality parameters comparable to those obtained from freshwater, cold-water fish, and mammalian species.

## 2. Results and Discussion

### 2.1. African Catfish By-Products Proximate Composition

The proximate composition of the ACBPs used in this study is presented in Table 1, along with a comparison to current research papers.

The observed variability, particularly the higher coefficient of variation for ash content (±1.7%), reflects inherent biological variability, including differences in bone content among individual fish by-products. The effects that can affect the AC composition are, above all, these: (i) Source consistency: All samples were sourced from a single aquaculture facility to minimize compositional variability arising from differences in diet, rearing conditions, gender, age, and genetic factors. (ii) RAS advantages: Fish reared in Recirculating Aquaculture Systems (RAS) exhibit more consistent composition compared to wild-caught specimens due to controlled feeding regimes and standardized environmental conditions. (iii) Anatomical variation: Different anatomical portions (heads, frames, viscera) inherently exhibit compositional differences, which contribute to the observed standard deviations in whole-by-product characterization. Compositional variability is a crucial consideration for industrial scalability.

This difference likely reflects the inclusion of bone and connective tissues in the current BP stream, which reduces the overall water and protein content while elevating the ash content compared to AC skin and meat. Although the chemical composition of AC tissues is influenced by multiple factors, including diet formulation, rearing conditions, age, and fish size, it is particularly affected in RASs, where feeding regimes are precisely controlled. Under such conditions, the nutritional profile of BPs—such as lipid, FA, protein, and ash content—tends to closely reflect the characteristics of the diet and production intensity. For instance, Shadieva et al. (2020) reported that increasing the proportion of plant-based ingredients in the feed led to decreased protein and increased fat levels in filets [39]. Similarly, Effiong and Yaro (2020) showed that the source of dietary lipids (e.g., plant vs. fish oil) significantly influenced the FA composition of muscle tissue [40]. In contrast, Chwastowska-Siwiecka et al. (2016) found that sex had no statistically significant effect on the proximate composition of the AC muscle [41].

### 2.2. Mass Balance of the Process

The defatting process yielded 18.17 ± 1.24% fish oil, while the production of fish protein hydrolysate and pigment resulted in 9.83 ± 2.14% and 1.72 ± 0.65%, respectively. The total gelatin yield obtained through four sequential extractions ranged from 25.3% to 37.8% (see Table 2), indicating good overall recovery efficiency. However, the first extraction fraction was consistently over-hydrolyzed, which prevented solid gel formation and suggested an excessive breakdown of collagen chains. Subsequent fractions exhibited improved gelling characteristics, with the second fraction consistently achieving the most favorable balance between yield and gel strength. The complete mass balance of the process reached 95.48 ± 2.95%, confirming high efficiency, as less than 5% of the initial raw material was lost during processing.

Statistical analysis (Table 3) of gelatin fraction yields showed strong model fits but varying levels of factor significance. For the first fraction yield (Y_1_), the model fit excellently (R^2^ = 97.77%), with no significant influence from any factor, although the temperature of the first extraction showed the strongest effect (*p* = 0.097). For the second fraction yield (Y_2_), the model demonstrated excellent explanatory power (R^2^ = 99.89%), with a statistically significant effect of the temperature of the first extraction (*p* = 0.026) and a near-significant influence of the temperature of the second extraction (*p* = 0.055). For the third fraction yield (Y_3_), the model fit remained strong (R^2^ = 98.73%), though none of the factors achieved statistical significance, with the temperature of the first extraction showing the strongest effect (*p* = 0.083), followed by the temperature of the second extraction (*p* = 0.224). The fourth fraction yield (Y_4_) exhibited similarly high explanatory power (R^2^ = 99.20%), yet statistical significance remained elusive, with the temperature of the second extraction demonstrating the most pronounced effect (*p* = 0.093) and a smaller contribution from the first extraction temperature (*p* = 0.205). The high R^2^ values across all fractions indicate that the models captured most of the variance, though the limited degrees of freedom restricted the ability to confirm statistical significance for individual factors. To visualize the influence of the factors on gelatin fraction yields, contour plots were prepared and are presented in Figure 1.

### 2.3. Life Cycle Assessment

The contribution analysis of individual processes involved in producing 1 kg of gelatin from catfish processing residues (Figure 2) reveals that the Pigment extraction and gelatin product drying processes dominate across all impact categories, consistently contributing more than 65% to each category. The Gelatin extraction process represents the second major contributor, accounting for approximately 30% across the impact categories. Since individual impact categories are expressed in different units and cannot be directly compared, normalization and weighting procedures were applied according to the Environmental Footprint (EF) 3.1 methodology. External normalization factors are expressed on a per capita basis per year, based on global impact category values [42]. The normalized results were subsequently multiplied by weighting factors that reflect the relative socioeconomic importance of each impact category [43]. The normalized and weighted results for the reference scenario (Figure 3) display the percentage contribution of each impact category to the total environmental impact.

Analysis of the weighted results (Table 4) identifies three impact categories with the largest contributions to total environmental impacts: Freshwater eutrophication, Fossils, and Climate change, each exceeding 10% of the total impact. The distribution of these three significant categories across individual processes (Figure 4) reveals remarkably similar patterns for all processes, except for Hexane recycling. This process introduces hexane, an organic, nonpolar solvent of fossil origin, used for fat extraction, which substantially increases the fossil resource consumption category for this specific process step. Examination of absolute weighted impact values (Table 5) confirms that the Pigment extraction and gelatin product drying process exhibits the highest values across all three significant impact categories. This is directly attributable to its substantial energy requirements, which represent the largest energy demand in the entire production cycle, primarily driven by the gelatin drying process.

Based on these initial findings, several hypotheses were formulated and tested through scenario simulations to analyze system sensitivity and identify potential improvement strategies. Given that Freshwater eutrophication, Fossils, and Climate change are heavily influenced by electricity production methods, the first alternative scenario replaced the Czech energy mix (approximately 50% fossil fuel-based) with the average European energy mix, which incorporates a significantly higher proportion of renewable energy sources. This scenario reflects the growing trend of companies adopting environmentally responsible practices, such as utilizing photovoltaic power plants. The second scenario focused on the efficiency of hexane recycling. In the reference scenario, 62% of hexane was recycled within the production cycle. To assess system sensitivity, two schemes were modeled: reducing the recycling rate by 20% to 42%, and increasing it by 20%. The higher recycling rate represents a realistic improvement under optimized recycling conditions. The third scenario examined the impact of water type. The reference scenario, which modeled anticipated industrial upscaling conditions, assumed tap water usage, requiring more than 2000 L per kg of gelatin produced. Over half of this volume is consumed in the Washing and drying process. Despite this substantial consumption, the Water scarcity impact category contributes only approximately 2.7% to total environmental impacts (Table 4). Since semi-industrial tests utilized deionized water, this scenario evaluated the substitution of tap water with deionized water to determine its environmental significance.

The transition to a greener energy mix produces substantial environmental benefits (Table 6). The three significant impact categories show considerable reductions: Freshwater eutrophication decreases by 67.85%, Fossils by 30.2%, and Climate change by 50.19%. Given that energy flow represents a critical input throughout the production cycle and energy consumption reduction is not practically feasible, adopting renewable energy sources emerges as the most effective strategy for reducing the environmental footprint of gelatin production from ACBPs. In contrast, the product system exhibits minimal sensitivity to variations in the hexane recycling rate. A 20% change in either direction results in weighted impact category changes ranging from only 0.01% to 1.2% (Table 7). Specifically, Freshwater eutrophication, Fossils, and Climate change show changes of 0.01%, 0.47%, and 0.16%, respectively, indicating that hexane recycling optimization offers limited potential for improving environmental impact. Similarly, water type substitution yields negligible changes in environmental impact, with percentage variations ranging from 0.001% to 1.779% across impact categories (Table 8). This demonstrates that the volume of water consumption is far more environmentally significant than its quality. Consequently, industrial-scale implementation should prioritize water conservation strategies. Water evaporating during processes, particularly Pigment extraction and gelatin products drying processes, can be recovered through condensation systems. For the Washing and drying process, which accounts for the highest water consumption, filtration or concentration technologies should be implemented to enable substantial regeneration of washing water.

Following current industry practices for environmental policy communication [44], the carbon footprint was calculated for all modeled scenarios (Figure 5). Results show remarkable similarity across scenarios, with one notable exception: the scenario employing renewable energy sources demonstrates significantly lower carbon emissions. Total electricity demand reaches 545.77 kWh per kg gelatin, with the Pigment extraction and gelatin products drying process consuming approximately 50% and Gelatin extraction accounting for 23% of total energy usage. These findings underscore that the implementation of energy-efficient equipment, particularly the adoption of renewable energy, represents the most effective pathway for reducing environmental impact. Water consumption constitutes the second major environmental hotspot, with approximately 2000 L required per kg of gelatin produced [31,32]. Rather than modifying water treatment processes, consumption reduction strategies should be prioritized. Implementing water recycling systems for evaporated water from the Pigment extraction and gelatin products drying process, as well as conditioning processes, offers immediate improvement potential. Most critically, the Washing and drying process requires appropriate wastewater treatment infrastructure, as the effluent contains high concentrations of organic nutrients that significantly contribute to aquatic environment eutrophication.

It should be emphasized that the LCA study was conducted for a semi-industrial scale of ACBP processing, which may not accurately reflect the size of material and energy flows in an up-scaled industrial process. It is likely that the flow of chemicals used, including enzymes, will increase in proportion to the processed batch of ACBP. However, on an industrial scale, less energy-intensive equipment can be used, and water consumption can also be lowered through the use of effective filtration and recycling methods. On the other hand, on an industrial scale, it may be necessary to transport waste for processing over longer distances, which would also require cooling or freezing.

Sampaio et al. [32] compare the LCA of a laboratory process for producing gelatin from tilapia skins with a possible process upscaled to pilot scale. They rely mainly on a comparison of energy consumption, which does not differ significantly in relation to the functional unit (120 g of gelatin). The authors state that the application of LCA at the laboratory stage of production can be beneficial for technology transfer to an industrial scale, as it can help estimate environmental impacts and guide upscaling to minimize the hotspots identified in the LCA study. In this way, product sustainability can be achieved.

When referring to other studies that present LCA of various gelatin production processes, whether on a laboratory scale from tilapia skins [32], single-step biocatalysis of bones [31], or industrial-scale production of pork gelatin [33], it should be noted that comparing them is not entirely straightforward. Each LCA study is valid for the geographical area for which it was developed, as well as for the time period from which the data used in the study originate. The methodologies used to calculate environmental impacts may also vary, as may the results when using different software. However, it can be generally observed that LCA studies focus primarily on energy and water consumption, the two most significant inputs into the gelatin production process. All studies emphasize the need for highly developed and efficient energy and water management to achieve sustainable gelatin production.

### 2.4. Physicochemical Properties of Gelatins

The functional properties of the gelatin obtained from ACBPs are summarized in Table 9 (gel strength and dynamic viscosity) and in Table 10 (melting and gelling point).

#### 2.4.1. Gel Strength and Dynamic Viscosity

Experiment 4 provided the highest overall yield; however, the resulting gel exhibited significantly lower gel strength (79.4 ± 3.1 Bloom). Conversely, the highest gel quality (168.8 ± 3.6 Bloom) was achieved in Experiment 2, which produced the third lowest yield, with a corresponding dynamic viscosity of 2.48 ± 0.02 mPa·s. Gel strength and dynamic viscosity measurements for Fractions 2, 3, and 4 (Table 9) revealed a strong positive linear relationship between these two rheological parameters. Pearson correlation analysis yielded coefficients of 0.955, 0.998, and 0.949 for the second, third, and fourth fractions, respectively, with all correlations being statistically significant (*p* < 0.05). Fractions that failed to form cohesive gels exhibited dynamic viscosities ranging from 0.72 ± 0.02 to 1.05 ± 0.05 mPa·s, whereas gel-forming fractions displayed substantially higher viscosities, ranging from 1.03 ± 0.03 to 3.62 ± 0.11 mPa·s.

When compared with published data, the present maximum gel strength of 168.8 ± 3.6 Bloom (viscosity 2.48 ± 0.02 mPa·s) is lower than the 234 ± 3 Bloom reported by Alfaro et al. (2014) for gelatin from AC skin obtained via acid–alkaline pretreatment [45], but falls within the medium Bloom range and matches their reported viscosity span of 2.05–2.85 mPa·s. Sanaei et al. (2013) found that AC bone gelatin yielded 230.25 Bloom and 4.64 mPa·s, while AC skin gelatin yielded 286.71 Bloom and 3.45 mPa·s using mixed acid and alkaline pretreatments—both higher than the best values recorded here [46]. See et al. (2013) reported that alkaline pretreatment with 0.01 N Ca(OH)_2_ resulted in the lowest gel strength (136.6 ± 0.89 Bloom; viscosity 1.72 ± 0.13 mPa·s), whereas a combined 0.2 N NaOH and 0.05 M acetic acid treatment produced the highest values (265.6 ± 4.22 Bloom; 3.62 ± 0.01 mPa·s). Notably, the upper viscosity range achieved in the present study for Experiment 2 Fraction 4 (3.62 ± 0.11 mPa·s) matches the best outcome reported by See et al., although corresponding Bloom values were lower [14].

#### 2.4.2. Melting and Gelling Point

The measured melting points of the gelatin fractions in this study ranged from 11.1 ± 0.2 °C to 28.3 ± 0.1 °C, while gelling points varied between 5.5 ± 0.2 °C and 16.4 ± 0.6 °C (Table 10). Several fractions, especially those obtained in Experiment 2, exhibited melting temperatures above 25 °C, aligning closely with values reported for AC skin gelatin (25.7 °C) by Alfaro et al. (2014) [45]. In contrast, the control sample displayed a lower melting point (19.7 ± 3.5 °C) and gelling point (10.6 ± 1.0 °C), indicating that fractionation processes can modify gelatin’s thermal transition behavior.

When compared with literature values, the observed melting points fall within the range reported for warm-water fish gelatins (typically 20–29 °C) but remain lower than those of mammalian gelatins (32.2–32.6 °C). Species such as grass carp, common carp, and rohu exhibit melting points of 28.1–29.1 °C and gelling points of 17.9–20.5 °C, while cold-water fish like cod display considerably lower values (8–10 °C) [46,47,48,49,50]. The relatively high melting point in certain fractions (e.g., Fraction 4) suggests a higher imino acid content or more compact triple-helix regions, as the thermal stability of gelatin has been positively correlated with proline and hydroxyproline levels [51,52].

Gelling points in this study (5.5–16.4 °C) are lower than those of tilapia (≈ 18 °C) [49] and most warm-water species [53]. These results confirm that AC gelatin fractions follow the general trend of fish gelatins having lower gelling and melting points than mammalian counterparts, due to their lower imino acid content [54]. Variations between fractions likely arise from differences in molecular weight distribution and peptide composition caused by the fractionation or extraction conditions.

Overall, the thermal transition data indicate that fractionation can yield AC fish gelatin fractions with melting points approaching those of commercial gelatins, offering potential advantages for applications requiring higher thermal stability compared to typical fish gelatin.

### 2.5. Composition and Properties of the Fish Oil

#### 2.5.1. Fatty Acid Composition

The FA composition (Table 11) analysis revealed a nutritionally balanced profile with SFAs comprising 30.85%, monounsaturated fatty acids (MUFAs) 40.52%, and PUFAs 28.65%, yielding an SFA:MUFA:PUFA ratio of approximately 11:14:10. Essential FAs that cannot be synthesized by the human body included LA (C18:2) at 14.60% and ALA (C18:3) at 2.02%, providing a total essential FA content of 16.62%. These health-promoting compounds provide significant protection against cardiac and vascular diseases, while also improving skin health [8,26]. These FA contents substantially exceeded values reported for farmed AC in previous studies, including 0.17% for farmed-raised to 11.65% for wild-raised specimens [55], 13.3% [36], 5.18% [56], and dietary-dependent variations ranging from 7.11% with coconut oil diet to 55.79% with sunflower oil diet [40], and extraction type dependent variation from 37.09% to 49.06% [24].

Conditionally essential long-chain PUFAs comprised arachidonic acid (0.67%), EPA (2.23%), and DHA (4.65%), with these bioactive compounds demonstrating therapeutic potential for treating rheumatoid arthritis, psoriasis, ulcerative colitis, asthma, Parkinson’s disease, osteoporosis, diabetes mellitus, cardiovascular events, cancers, and depression, while providing crucial support for brain and retinal functions [26,57]. The arachidonic acid content aligned well with literature values of 0.6% [36], 0.1% [25], 1.77% from AC viscera [56], and dietary-dependent ranges of 0.01–0.67% [40] and 0.02-5.74% with higher yields achieved through the kiln smoking extraction method [24]. EPA content proved comparable to commonly used salmon oil levels (1.8–4.2%) [58] and significantly exceeded values reported by Habib and Sarkar (2016) (0.8%) and Effiong and Yaro (2020) (0.02–1.18%) [25,40]. DHA demonstrated relatively high content compared to AC oils reported by Eke-Ejiofor et Ansa (2018) (0.95–1.06%), Habib and Sarkar (2016) (2.48%), Osibona et al. (2009) (3.0%), Effiong and Yaro (2020) (0.03–0.97%), and Sathivel et al. (2003) (0.62%) though remaining lower than salmon oil DHA levels (10.8–12.8%) [24,25,36,40,56].

Among SFAs, PA dominated at 20.45%, corresponding well with literature reports of 22.0% [36], 13.92% from AC viscera [56], 20.0% [25], while significantly exceeding other reported values ranging from 1.37% [24] through 2.18% [55] and 6.63% [23] to 8.03%, while palm oil-fed specimens had an exceptionally high PA content, 23.4% [40]. SA content (6.65%) fell within the middle range of published literature values spanning from 0.1% [25], 0.26% [23], 0.62-0.99% [55], 0.84–7.78% [24], 4.24–8.28% [40], 8.1% [36], and 11.97% from AC viscera [56], with both PAs and SAs serving as precursors for oleic acid biosynthesis [59]. Oleic acid (29.08%), the dominant MUFA, exceeded the range (17.69–23.73%) typically reported for catfish oils [25,55] and provides documented beneficial effects on pancreatic and liver function [59]. Myristic acid appeared in relatively low proportion (2.28%) yet exceeded levels reported for farmed (0.15%) and wild AC (0.67%), contributing positively to cardiovascular health [55]. Additional nutritionally significant MUFAs included palmitoleic acid (4.19%), found in concentrations similar to previous reports by Habib and Sarkar (2016) (3.9%), Osibona et al. (2009) (3.6%) [25,36], and within ranges reported by Eke-Ejiofor et Ansa (2018) (2.29–10.5%) and Effiong and Yaro (2020) (0.4–7.47%) under different dietary conditions [24,40], though lower than Sathivel et al. (2003) (8.67%) [56]. This FA demonstrates particular importance given its established links to improved insulin sensitivity and reduced diabetes risk [60]. Gadoleic acid, a short-chain MUFA, showed comparable results (2.6%) to Osibona et al. (2009) (2.5%) [36] while substantially exceeding values reported by Habib and Sarkar (2016) (0.03%) and Effiong and Yaro (2020) (0.05–0.51%) [25,40].

Overall, the studied oil demonstrates a nutritionally advantageous profile characterized by elevated MUFA content, favorable PUFA composition, and comparatively high levels of health-promoting FAs including LA, oleic acid, EPA, and DHA. These comprehensive findings strongly suggest that ACBP oil could serve as a valuable raw material for nutraceutical and functional food applications, while simultaneously supporting sustainable valorization strategies within circular economy frameworks.

#### 2.5.2. Peroxide Value

The peroxide value (PV) serves as a critical indicator of lipid oxidation and oil quality. Fresh edible oils typically exhibit PVs below 10 mEq O_2_/kg, while rancid oils generally exceed 30 mEq O_2_/kg; values above 100 mEq O_2_/kg have been associated with food poisoning [61]. Gotoh and Wada (2006) emphasized that monitoring PV is essential for both food quality assurance and safety, as oxidation reactions produce compounds that adversely affect taste, shelf life, and freshness [62].

The PV of freshly prepared fish oil samples was significantly lower (16.88 ± 0.84 mEq O_2_/kg) than that of the six-month-old samples, with the aged samples exhibiting approximately four times the value (75.76 ± 0.58 mEq O_2_/kg). The aged fish oil was stored in a refrigerator (7 ± 1 °C). This difference underscores the sensitivity of fish oil to autooxidation, a process wherein oxygen reacts with unsaturated FAs to form primary oxidation products—peroxides.

Our findings align with previously reported values for AC oil. Adetuyi et al. (2013) obtained 16.60 ± 0.6 mEq O_2_/kg from AC viscera oil [63], while Ningrum et al. (2023) reported PVs of 5.15 ± 0.146 and 4.15 ± 0.114 mEq O_2_/kg for AC head and flesh oils, respectively [64]. The wide variation in reported PVs can be attributed to differences in extraction methods, as demonstrated by Sathivel et al. (2009), who found PV ranging from 4.9 ± 0.46 to 36.12 ± 0.18 mEq O_2_/kg depending on the extraction technique employed for AC viscera oil [65].

Storage conditions significantly influence PV development. Famurewa et al. (2017) monitored AC oil over six weeks under various temperature conditions: at room temperature, PV increased from 4.60 ± 0.2 to 7.37 ± 0.15 mEq O_2_/kg; at 40 ± 4 °C, values ranged from 6.3 ± 0.3 to 7.55 ± 0.16 mEq O_2_/kg; while at −6 °C, PV increased more substantially from 6.3 ± 0.3 to 16.02 ± 0.42 mEq O_2_/kg [66]. Ming (2023) further confirmed the temperature dependency of PV in AC oil, demonstrating that elevated temperatures accelerate oil degradation [67]. These observations highlight the importance of maintaining appropriate storage conditions, including temperature control and light protection, to minimize oxidative deterioration and preserve oil quality.

### 2.6. The Innovation of the Study and Practical Implications

This study demonstrates the comprehensive valorization potential of ACBPs, addressing critical challenges in fish processing waste management and sustainable resource utilization. The significance of this work lies primarily in the application of a unique biotechnological approach for collagen conditioning using proteolytic enzymes, which disrupts the quaternary structure of collagen in a manner that enables subsequent extraction steps to produce high-quality gelatins. Unlike conventional methods that employ weak acid or alkaline solutions to disrupt collagen structure, this enzymatic approach is significantly more environmentally benign.

Gelatin extraction from ACBPs yields 25.3–37.8%, with specific fractions exhibiting gel strengths up to 168.8 ± 3.6 Bloom and thermal transition properties (melting points 15–26 °C, gelling points 6–14 °C) that approach several commercial gelatin specifications. These functional characteristics establish AC gelatin as a viable alternative to mammalian sources, particularly suited for applications requiring moderate gel strength and lower thermal stability thresholds.

The co-extracted fish oil presents significant nutritional value, with a balanced FA profile comprising 40.52% MUFAs and 28.65% PUFAs. Notably, the oil contains therapeutically relevant omega-3 FAs (EPA: 2.23%, DHA: 4.65%) and essential FAs (16.62%), making it suitable for nutraceutical and functional food applications. However, proper storage conditions are critical, as peroxide value measurements demonstrate that six-month refrigerated storage without light protection increases oxidation approximately fourfold (from 16.88 ± 0.84 to 75.76 ± 0.58 mEq O_2_/kg), emphasizing the need for appropriate preservation strategies, including light protection and temperature control.

The LCA reveals that environmental optimization hinges primarily on energy management, with the Pigment extraction and gelatin drying process contributing over 65% to environmental impacts. Transitioning to renewable energy sources demonstrates substantial reductions in key impact categories: freshwater eutrophication (−67.85%), climate change (−50.19%), and fossil resource consumption (−30.2%). These findings provide actionable guidance for industrial implementation, indicating that the sustainable production of gelatin and oil from ACBPs is achievable through the adoption of renewable energy and water conservation strategies, thereby supporting circular economy principles in aquaculture processing industries.

To determine the optimal conditions for gelatin processing, both gelatin yield, an industrially important factor, and gel strength, which dictates product applicability, must be considered. Experiment 2 (35 °C first fraction, 60 °C second fraction) achieved the highest gel strength (168.8 ± 3.6 Bloom) while maintaining reasonable yield, whereas Experiment 4 (45 °C first fraction, 60 °C second fraction), despite providing superior overall yield, exhibited significantly reduced gel quality (79.4 ± 3.1 Bloom). The second gelatin fraction consistently demonstrated the most favorable balance between yield and functional properties across all experimental conditions, making it the preferred target for gelatin recovery from fish BPs.

Our process incorporates several factors that minimize the risk of microbial contamination. (i) Low-temperature enzymatic conditioning: Processing at 10 °C for 3 h significantly inhibits microbial growth during enzyme treatment. (ii) Thermal inactivation: Subsequent heating to 85 °C for 4 min provides effective thermal inactivation of potential contaminants. (iii) Food-grade enzyme: Protamex^®^ (Novozymes, Copenhagen, Denmark) meets JECFA/FCC purity standards, ensuring the enzyme itself does not introduce microbial contamination. (iv) Final sterilization: In industrial gelatin production, sterilization is a standard final processing step.

### 2.7. Limitations of the Study and Future Perspectives

The primary limitation of the presented study concerns the management of waste maceration liquor generated after demineralization of ACBP skeletons. Adopting a zero-waste approach, we recommend processing the maceration liquor according to the procedure described in our previous publication [68], wherein precipitation with Ca(OH)_2_ produces feed-grade dicalcium phosphate dihydrate (CaHPO_4_·2H_2_O) suitable for crop and agricultural production. Another limitation is related to the cost-effective solvent extraction method used for recovering fish oil. Although hexane extraction is economically viable, rigorous testing must be conducted to ensure that the extracted fish oil contains negligible hexane residues, thereby guaranteeing its safety for food, pharmaceutical, or nutraceutical applications. A further limitation involves identifying suitable applications for two BPs: the solid cake remaining after extraction of the fourth gelatin fraction, which contains undecomposed collagen residue, and the fish pigment extract. From an industrial perspective, a further and perhaps the most critical limitation is the scalability of the entire process. Scaling up is challenging due to the need to design and implement properly functioning equipment, as well as to test its operational capacities and performance under industrial conditions. Addressing this limitation will be essential for translating laboratory-scale results into viable industrial applications.

Future research should prioritize detailed compositional characterization of both the solid cake and pigment extract as a critical prerequisite for identifying optimal valorization pathways. The conversion of solid cake into higher-value-added products presents several promising strategies. Enzymatic hydrolysis using proteolytic enzymes could produce collagen hydrolysates suitable for application as nutritional supplements or animal feed components. Alternatively, fermentation-based bioprocessing may yield food-grade additives with functional properties. Regarding the fish pigment extract, a systematic evaluation of its dyeing capacity, stability, and bioactivity is warranted to assess its potential applications across various industries. Beyond BP valorization, process optimization is essential for enhancing sustainability. The LCA identifies energy and water consumption as primary environmental hotspots requiring targeted mitigation strategies.

## 3. Conclusions

The results of the proposed ACBP processing technology confirmed the hypothetical assumption of producing high-quality gelatin under optimized conditions with the lowest possible environmental impact. The presented comprehensive technology for processing unused ACBP skeletons is characterized by a high efficiency of collagen conversion into collagen products—gelatins and hydrolysates—with overall yields exceeding those typically achieved in the processing of BPs from marine and freshwater fish. The prepared gelatins exhibit excellent gelling and surface properties, as well as low residual mineral content, making them suitable for the development of high-value-added matrices, such as in pharmaceuticals (soft and hard capsules), the food industry, or cosmetics. Fish oil with a high content of essential FAs, along with collagen hydrolysate, can be primarily used in the production of nutritional food supplements.

The LCA confirms that gelatin production from ACBPs is both energy-intensive and water-intensive. The application of the Environmental Footprint methodology reveals two primary environmental hotspots: electricity consumption (545.77 kWh/kg of gelatin) and water consumption (approximately 2000 L/kg of gelatin). Scenario analysis demonstrates that transitioning to renewable energy sources produces the most substantial environmental benefit, reducing the carbon footprint by up to 50% while significantly decreasing freshwater eutrophication and fossil resource depletion. Water conservation through recycling and advanced treatment technologies represents the second critical improvement opportunity, particularly for managing organic nutrient-rich wastewater from the Washing and drying process. These findings provide clear guidance for industrial-scale implementation of sustainable gelatin production from aquaculture processing residues.

Our study provides measurable indicators of circular bioeconomy alignment through two primary approaches. (i) Life Cycle Assessment (LCA): We conducted a comprehensive LCA analysis to quantify the environmental impacts and sustainability metrics of the valorization process, providing objective measurements of resource efficiency and environmental performance. (ii) By-product utilization: Our process design demonstrates systematic valorization of all by-products, with minimal waste generation at the end of the sequential extraction pathway. This is evidenced by the fractionation and utilization of multiple product streams (gelatin fractions, fish oil, mineral components) from the initial aquaculture waste material.

## 4. Materials and Methods

### 4.1. African Catfish By-Products and Their Proximate Composition

Samples of ACBPs, consisting of skeletons with residual soft tissue and skin, were obtained from Tilapia s.r.o. (Radenín-Hroby, Czech Republic), a commercial recirculating aquaculture system (RAS) facility specializing in the production and processing of AC. The fish were reared in temperature-controlled tanks at approximately 26 °C and harvested at an average body weight of 1.2–1.3 kg and age of 10–12 months (matured fish). They were fed a commercial extruded diet EFICO Alpha 717 from BioMar A/S (Brande, Denmark). ACBPs were collected immediately after manual fileting under hygienic conditions. The remaining skeletons with attached soft tissue were processed using a belt-type mechanical separator. The resulting material was stored at 4 °C and transported to the laboratory for further processing within 24 h.

The ACBPs were subjected to initial compositional analysis: moisture content was determined gravimetrically [69], ash content was measured after incineration [70], lipid content was assessed via Soxhlet extraction [71], and protein content was calculated from nitrogen content determined by the Kjeldahl method [72].

### 4.2. Appliances and Chemicals

The chemicals used were NaCl, NaOH, HCl, petroleum ether, hexane, and ethanol; all chemicals were of analytical grade and purchased from Lach-Ner s.r.o. (Neratovice, Czech Republic). Protamex^®^, an endopeptidase that cleaves peptide bonds within the protein structure, generating peptides of higher molecular weight, was obtained from Novozymes (Copenhagen, Denmark). The enzyme exhibits optimal activity at a pH range of 5.5–7.5 and temperatures between 35 and 60 °C. Protamex^®^ meets the recommended purity specifications for food-grade enzymes established by the Joint FAO/WHO Expert Committee on Food Additives (JECFA) and the Food Chemicals Codex (FCC). The compatibility of Protamex^®^ enzyme with fish species was demonstrated in previous research by the authors, which involved the extraction of *Cyprinus carpio* skeleton gelatin [68].

### 4.3. Experimental Design and Statistical Analysis

The experiments were designed using a factorial approach with two factors. Factor A corresponded to the extraction temperature of the first gelatin fraction (35 and 45 °C), while Factor B corresponded to the extraction temperature of the second fraction (50 and 60 °C). This design enables the identification of the most influential process factors and the determination of their optimal values. All analytical determinations were conducted in triplicate. Data are expressed as mean ± standard deviation, and results with *p* < 0.05 were considered significant. Data were statistically analyzed using Minitab 17 statistical software (Fujitsu Ltd., Tokyo, Japan). The means were compared using an ANOVA test.

### 4.4. Methodology of the Raw Material Processing

Figure 6 illustrates the sequential processing steps for converting ACBPs into gelatin, hydrolysate, oil, and pigments.

Bones were minced (Braher P22/82, 13 mm plate, double-sided knife; Braher, San Sebastian, Spain), vacuum-packed in 1 kg portions (Henkelman Vacuum Systems, CK ‘s-Hertogenbosch, The Netherlands), frozen at −85 ± 1 °C for 12 h, then stored at −18 ± 1 °C. Prior to processing, the material was thawed at 5.0 ± 1.0 °C for 24 h. Purification involved removing non-collagenous proteins, fat, and minerals through successive treatments: a cold water rinse, followed by 0.2 M NaCl (1:6 w/v, 1.5 h) to eliminate globulins, and then four 0.03 M NaOH treatments (1:6 w/v, 45 min each) with intermittent stirring and cold-water rinses. The purified material was dried at 35 ± 1 °C for 24–36 h using a Memmert ULP 400 (Memmert GmbH, Büchenbach, Germany). Defatting was performed using a 1:6 (w/v) hexane solution with agitation (LT 43 shaker, Nedform Ltd., Valašské Meziříčí, Czech Republic) for 1.5–2 days at room temperature, with two solvent changes. After drying, the material was milled to approximately 3 mm particles (IKA A 10, IKA-Werke GmbH, Staufen, Germany) and stored sealed at 22 ± 1 °C. Solvent was recovered by vacuum evaporation (Heidolph Laborota 4010 Digital, Heidolph Instruments GmbH & Co. KG, Schwabach, Germany), and oil yield was determined gravimetrically. Demineralization used 1.0% HCl (1:10 *w*/*v*) at 10 ± 1 °C for 48 h with acid replacement after 24 h (LT 43 shaker, Nedform Ltd., Valašské Meziříčí, Czech Republic; Lovibond Thermostatically Controlled Cabinet, Tintometer GmbH, Dortmund, Germany). Material was rinsed, dried at 35 ± 1 °C for 24–36 h (Memmert ULP 400, Memmert GmbH, Büchenbach, Germany), and milled to approximately 2 mm particles (IKA A 10, IKA-Werke GmbH, Staufen, Germany). Enzyme conditioning mixed 60.0 g dried collagen with 600 mL distilled water and 0.1% Protamex^®^ (0.056 g, based on 56.209 g dry matter of the 60.0 g sample). pH was adjusted to 6.0–7.0 using 20% NaOH and monitored manually throughout the process (WTW pH 526, WTW GmbH, Oberbayern, Germany). The mixture was incubated at 10 ± 1 °C for 3 h with continuous shaking using an LT 43 shaker (Nedform Ltd., Valašské Meziříčí, Czech Republic) and a Lovibond Thermostatically Controlled Cabinet (Tintometer GmbH, Dortmund, Germany). After enzymatic treatment, the hydrolysate was filtered and dried at 70 ± 1 °C for 24 h. The enzyme-treated material was rinsed for 5 min to remove any residual enzyme before further processing. Sequential gelatin extractions used hot water (1:10 *w*/*v*) with magnetic stirring (IKA LABORTECHNIK RCT BASIC, Staufen, Germany): 35–45 ± 1 °C (first changing factor: factor A) for 20 min, 50–60 ± 1 °C (second changing factor: factor B) for 25 min, 70 ± 1 °C for 20 min, and 90 ± 1 °C for 30 min. After the first two extractions, rapid heating to 85 ± 1 °C for 4 min inactivated residual enzyme. Gelatin solutions were filtered, centrifuged for 5 min at 8000 rpm (Rotina 35, Hettich Zentrifugen, Tuttlingen, Germany) for pigment obtainment, dried stepwise (40 ± 1 °C, 20 h), and ground to 1 mm.

Specific temperature pairings were selected for the following reasons. The initial extraction temperature at 35–45 °C in the first extraction step was selected to solubilize loosely bound collagen molecules and telopeptide-rich fractions without inducing full collagen denaturation. This pre-treatment temperature is below the typical denaturation range of fish collagen, allowing selective extraction of the most thermally labile components while preserving the integrity of more stable collagen structures for subsequent fractions. The 50–60 °C in the second extraction step targets the main collagen denaturation temperature of fish species, which ranges from 50 to 60 °C depending on species and habitat temperature. This temperature ensures complete conversion of native collagen to gelatin while maintaining optimal molecular weight distribution. Literature consistently identifies this range as producing the highest quality fish gelatin with desirable gel strength and functional properties. The third extraction step, performed at 70 °C, was selected to extract residual collagen with higher thermal stability and cross-linked structures that resist denaturation at lower temperatures. It represents a practical upper limit before excessive peptide bond hydrolysis compromises gelatin quality. Rationale for 90 °C (fourth gelatin fraction) was included as an exploratory condition to assess whether any extractable material remained and to quantify the quality-yield trade-off at elevated temperatures. As noted in our results, this fraction consistently yielded minimal quantities with inferior quality characteristics.

### 4.5. Life Cycle Assessment of the Process

The environmental impact of fish gelatin production was assessed using LCA according to ISO 14040 and ISO 14044 standards [73,74]. A cradle-to-gate approach was adopted, with a functional unit of 1 kg of gelatin, encompassing both raw material processing and production stages (Figure 7). Fish processing waste from the farm was considered feedstock entering with zero environmental burden and no transport requirements. The system boundary ended at the factory gate, excluding packaging, distribution, and end-of-life stages; BPs (fats, minerals, pigments, collagen hydrolysates) were allocated physically based on mass ratios. Hexane defatting was modeled with a 62% recycling efficiency, using the Czech Republic’s energy mix for electricity, and assuming tap water for water consumption. The Protamex^®^ enzyme input was excluded due to its negligible mass. Environmental impacts were calculated using Umberto 11 software (iPoint-systems) with ecoinvent 3.11 database and the Product Environmental Footprint (EF 3.1) method [75], assessing climate change, human health, ecosystem quality, and resource depletion. Foreground data were obtained from laboratory measurements simulating semi-commercial operation (February–May 2025), while background data for chemicals and electricity were sourced from ecoinvent 3.11 [76,77] using Czech, European, or global averages as appropriate (Appendix A).

### 4.6. Properties of the Prepared Products

#### 4.6.1. Fish Gelatin Properties

Gel strength and viscosity were determined according to the standard testing methods for edible gelatins [78]. Gel strength was measured using a Stevens LFRA Texture Analyzer (Leonard Farnell and Co., Ltd., Norfolk, UK) on 6.67% gelatin gels matured at 10 ± 1 °C for 17–18 h and then compressed by a 4 mm by 12.7 mm diameter flat-faced cylindrical Teflon plunge. The surface quality was assessed using the parameters specified in the standard [79]. Dynamic viscosity was determined by measuring the flow time at 60 ± 0.5 °C using a Haake P5 Circulating Bath with a Thermo C10 Controller (Thermo Fisher Scientific, Waltham, MA, USA) and a Ubbelohde viscometer.

Yield calculations were based on the percentage of gelatin obtained relative to the weight of the purified raw material. For each gelatin fraction, yield was calculated as the mass of gelatin produced divided by the initial mass of purified raw material and expressed as a percentage (Equation (1)). Yields for both the hydrolysate and pigment fractions were determined using the same method (Equations (2) and (3)). The yield of the residual solid fraction, referred to as the “solid cake,” was also calculated based on the mass remaining after each gelatin extraction, relative to the starting raw material (Equation (4)). Total gelatin yield is calculated as the sum of individual gelatin yields (Equation (5)). The mass balance error (MBE) was determined by summing all individual yields and subtracting the total from 100% (Equation (6)). Any discrepancy in the MBE likely results from trace amounts of raw material or gelatin solution lost during processing equipment or container handling.(1)$$Y_{G}=\left(\frac{m_{G}}{m_{0}}\right)\times 100$$(2)$$Y_{H}=\left(\frac{m_{H}}{m_{0}}\right)\times 100$$(3)$$Y_{P}=\left(\frac{m_{P}}{m_{0}}\right)\times 100$$(4)$$Y_{SC}=\left(\frac{m_{SC}}{m_{0}}\right)\times 100$$(5)$$\sum Y_{G}=Y_{G1}+Y_{G2}+Y_{G3}+Y_{G4}$$(6)$$MBE=\left[100-\left(Y_{H}+Y_{P}+\sum Y_{G}\right)\right]$$
where $Y_{H}$ is the yield of hydrolysate [%], $Y_{G}$ is the yield of the gelatin fractions [%], $Y_{G1}$ is the yield of the first gelatin fraction [%], $Y_{G2}$ is the yield of the second gelatin fraction [%], $Y_{G3}$ is the yield of the third gelatin fraction [%], $Y_{G4}$ is the yield of the fourth gelatin fraction [%], $Y_{P}$ is the yield of pigment [%], $Y_{SC}$ is the solid cake (solid residual after extraction) [%], $Y_{G}$ is total gelatin yield [%], $m_{0}$ is the weight of the purified raw material [g], $m_{H}$ is the weight of the hydrolysate [g], $m_{G}$ is the weight of gelatins [g], $m_{P}$ is the weight of pigment [g], $m_{SC}$ is the weight of the solid cake [g], and the *MBE* is the mass balance error.

The melting point was measured using the capillary tube method with 6.67% gelatin gels described by Chen et al. (2022) [80]. 2–4 mm diameter capillaries were filled with the prepared gelatin samples to a depth of 0.5–1 cm. Each capillary was marked to indicate gel height and stored in a refrigerator to solidify. For the melting point measurement, a heated water bath with magnetic stirring was set up. A capillary was positioned alongside a temperature probe inside a test tube submerged in distilled water. Heating was initiated at a rate of 1.5 °C/min, and the temperature at which the gel column began to rise was recorded as the melting point. Each sample was analyzed in triplicate. The gelling point was measured using the same 6.67% gelatin gel samples described in the article. A melted gelatin sample was poured into a 10 mm diameter glass test tube and allowed to cool while submerged in chilled water. The sample temperature was reduced at a rate of 3 °C/min by adding ice water. A temperature probe monitored the cooling process. 0.1 g of metal balls were dropped into the sample at 1 °C intervals. The temperature at which the ball no longer sank marked the gelling point.

#### 4.6.2. Fish Oil Fatty Acid Profile

FA profiles were determined following ISO standards with minor modifications [81,82,83,84]. Samples (0.4 g of raw material) were subjected to saponification using a microwave reaction system with 5 mL of 2.8% methanolic KOH solution. The heating program consisted of a ramp to 90 °C (at a rate of 5 °C/min), followed by a 10 min hold at 90 °C. The resulting mixture was methylated using 15 mL of methanolic acetyl chloride with a temperature program of a 5 min ramp to 120 °C followed by a 6 min hold. After cooling, the FAs were extracted with 10 mL of heptane and then salted out using a saturated sodium chloride solution. The organic layer was separated, filtered through 0.22 μm filters, and dried over anhydrous sodium sulfate. Fatty acid methyl esters (FAMEs) were analyzed using a Shimadzu GCMS-TQ8040 NX gas chromatograph coupled with triple quadrupole mass spectrometer (Shimadzu, Kyoto, Japan). Data were evaluated using GCMS solution software, Version 4.52 (Shimadzu, Kyoto, Japan). Parameters of the analysis are shown in Appendix A.

#### 4.6.3. Fish Oil Peroxide Value

The PV was determined by iodometric titration using a standardized 0.01 M sodium thiosulfate solution according to the ISO standard with minor modification [85]. The Na_2_S_2_O_3_ solution was prepared by dissolving 0.625 g of Na_2_S_2_O_3_·5H_2_O in 250 mL of distilled water and standardized against a primary 0.01N potassium dichromate (K_2_Cr_2_O_7_) solution using potassium iodide and hydrochloric acid. For the PV determination, 1.0–1.5 g of fat was accurately weighed into a 250 mL Erlenmeyer flask, dissolved in 25 mL of chloroform–acetic acid (1:1, *v*/*v*), and treated with 1 mL KI. After 5 min in the dark, 75 mL of distilled water and 2 mL of starch indicator were added, and the liberated iodine was titrated with the standardized Na_2_S_2_O_3_ solution until the color disappeared. A blank was run in parallel, and the PV was calculated as shown in Equation (7).(7)$$PV=\frac{1000\times M\times (a-b)}{n}\times 2$$
where *PV* is peroxide value [mEq O_2_/kg], *M* is the exact concentration of the prepared Na_2_S_2_O_3_ solution, *a* is the volume of 0.01 M Na_2_S_2_O_3_ required to titrate the sample [ml], *b* is the volume of 0.01 M Na_2_S_2_O_3_ required to titrate the blank [ml], and *n* is the mass of oil added to the sample [g].

European Patent No. EP3707215. “Biotechnology-based production of food gelatine from poultry by-products.” Granted June 25, 2025. Proprietor: Tomas Bata University in Zlín, Nám. T. G. Masaryka 5555, 76001 Zlín, Czech Republic. European Patent Office, Munich.

## Figures and Tables

**Figure 1 gels-12-00045-f001:**
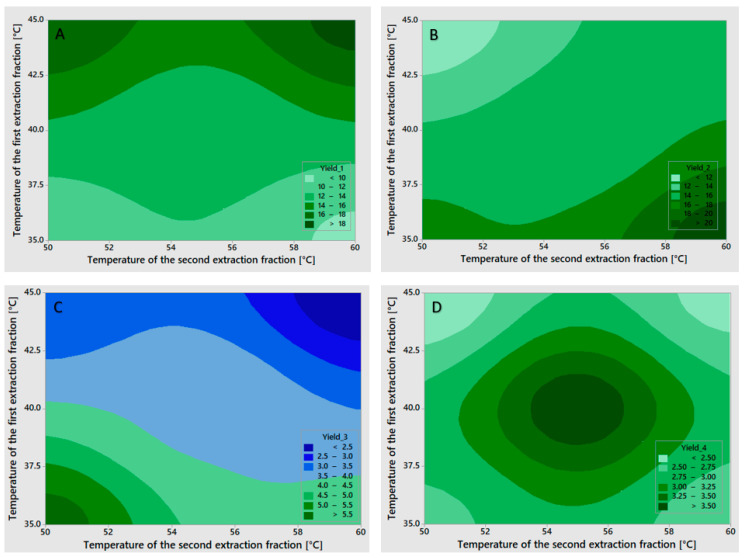
The influence of the temperature of the first and the second extraction on gelatin yields: (**A**) the yield of the first gelatin fraction; (**B**) the yield of the second gelatin fraction; (**C**) the yield of the third gelatin fraction; (**D**) the yield of the fourth gelatin fraction.

**Figure 2 gels-12-00045-f002:**
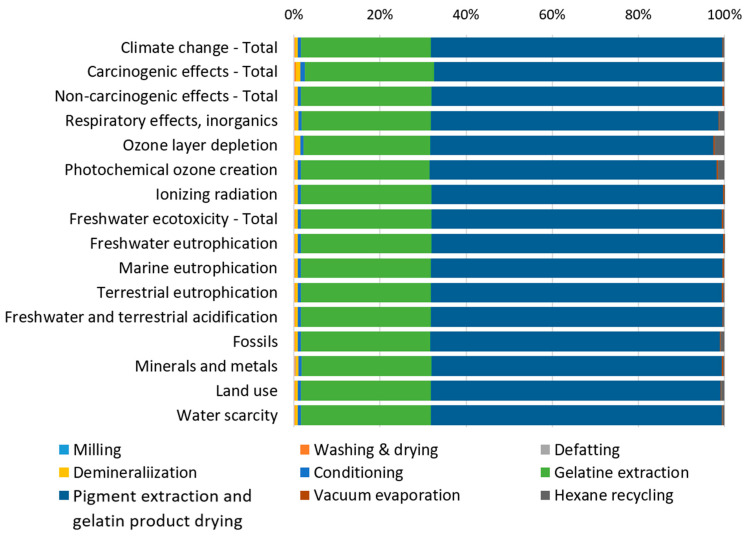
Percentage contribution of individual processes to environmental impacts by impact category.

**Figure 3 gels-12-00045-f003:**
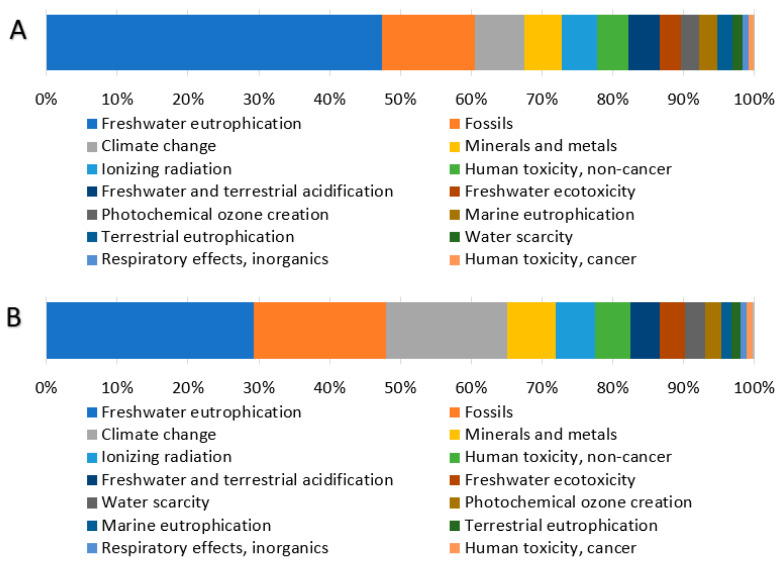
Percentage contribution of individual impact categories to normalized (**A**) and weighted (**B**) results.

**Figure 4 gels-12-00045-f004:**
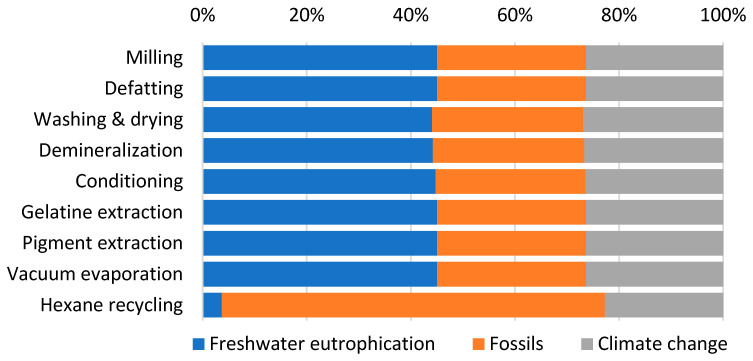
Percentage contributions of the gelatin individual production processes to three significant impact categories (weighted results).

**Figure 5 gels-12-00045-f005:**
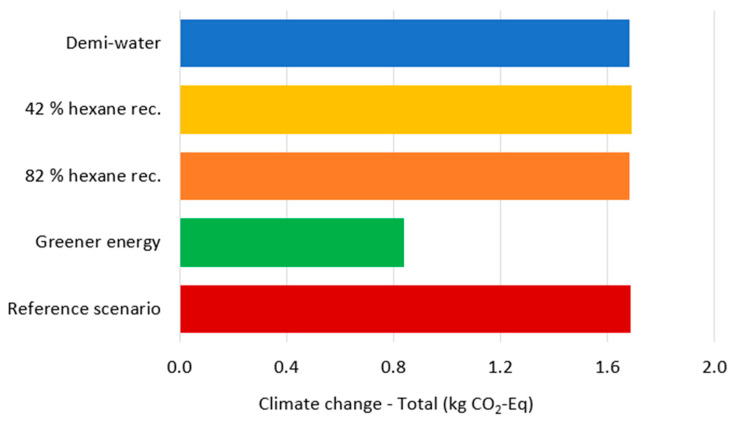
Comparison of the total values of the carbon footprint of individual scenarios.

**Figure 6 gels-12-00045-f006:**
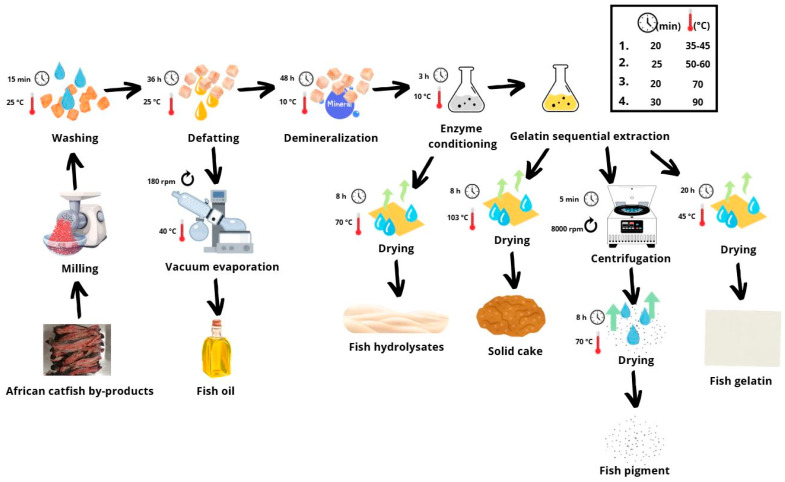
Flow chart of the integrated processing of ACBP skeletons into gelatin, hydrolysate, oil, and pigments.

**Figure 7 gels-12-00045-f007:**
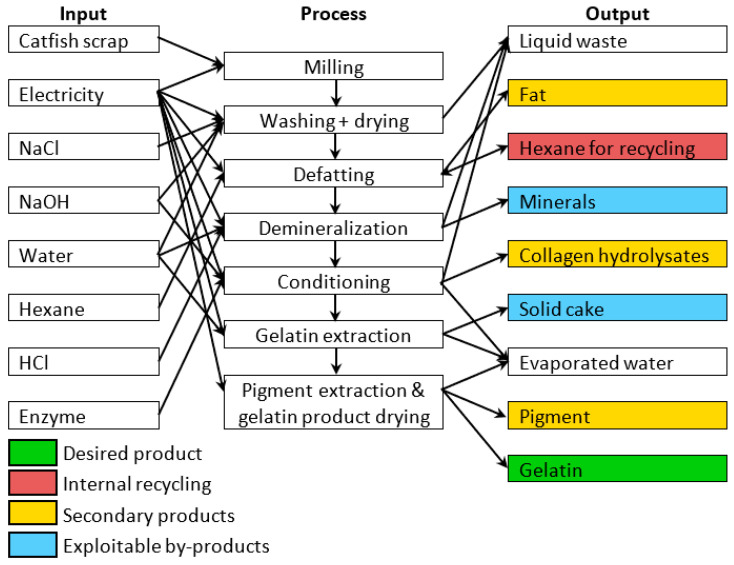
The LCA system boundaries.

**Table 1 gels-12-00045-t001:** Proximate composition of different AC sources compared with our results.

Source	Moisture [%]	Protein [%]	Lipid [%]	Ash [%]
ACBPs [this study]	65.80 ± 1.5	15.40 ± 1.2	10.00 ± 1.1	5.20 ± 1.7
AC muscle and skin [9]	71.30 ± 0.15	19.03 ± 0.46	8.10 ± 0.09	1.05 ± 0.14
AC muscle and skin [35]	76.27 ± 1.07	18.48 ± 0.91	11.00 ± 1.21	8.03 ± 0.88
AC muscle and skin [36]	74.30 ± 3.7	18.80 ± 1.9	9.30 ± 0.3	1.20 ± 0.5
AC muscle and skin [37]	77.20	17.12 ± 0.12	3.39 ± 0.78	1.53 ± 0.12
AC muscle and skin [38]	71.85 ± 0.07	19.51 ± 0.18	14.28 ± 0.19	3.06 ± 0.04

**Table 2 gels-12-00045-t002:** Gelatin yield for each fraction and total yield under different experimental conditions. The control experiment was conducted without enzyme addition during the conditioning step.

Experiment	Factors		Yield of Gelatin [%]				
	Temperature of the First Extraction [°C]	Temperature of the Second Extraction [°C]	Fractions				Σ Y_G_
			*Y* _*G*1_	*Y* _*G*2_	*Y* _*G*3_	*Y* _*G*4_	
1	35	50	10.80 ± 0.90	16.34 ± 0.92	5.72 ± 2.56	2.68 ± 0.01	35.55 ± 2.57
2	35	60	9.64 ± 1.06	20.95 ± 0.74	4.10 ± 0.69	2.57 ± 0.44	34.76 ± 1.95
3	45	50	17.35 ± 1.43	10.52 ± 0.19	3.07 ± 0.42	2.28 ± 0.29	33.22 ± 2.34
4	45	60	18.58 ± 0.25	14.69 ± 2.50	2.12 ± 0.66	2.39 ± 0.50	37.79 ± 2.41
5	40	55	12.40 ± 0.73	14.46 ± 1.17	3.93 ± 0.44	3.72 ± 0.29	34.50 ± 0.99
Control	40	55	6.16 ± 0. 47	7.15 ± 0.57	6.16 ± 0.50	5.82 ± 0.14	25.29 ± 0.71

**Table 3 gels-12-00045-t003:** Analysis of variance of the experimental design for gelatin yields.

	Degree of Freedom	Sum of Squares	Mean Squares	F-Value	*p*-Value
Response: 1st gelatin fraction yield; R^2^ = 97.77%					
Regression	3	63.152	21.051	14.62	0.189
Factor A (Temp. of 1st fraction)	1	60.84	60.84	42.25	0.097
Factor B (Temp. of 2nd fraction)	2	2.312	1.156	0.8	0.62
Error	1	1.44	1.44		
Total	4	64.592	64.592		
Response: 2nd gelatin fraction yield; R^2^ = 99.89%					
Regression	3	57.4175	19.1392	306.23	0.042
Factor A (Temp. of 1st fraction)	1	36.6025	36.6025	585.64	0.026 *
Factor B (Temp. of 2nd fraction)	2	20.815	10.4075	166.52	0.055
Error	1	0.0625	0.0625		
Total	4	57.48			
Response: 3rd gelatin fraction yield; R^2^ = 98.73%					
Regression	3	6.998	2.33267	25.92	0.143
Factor A (Temp. of 1st fraction)	1	5.29	5.29	58.78	0.083
Factor B (Temp. of 2nd fraction)	2	1.708	0.854	9.49	0.224
Error	1	0.09	0.09		
Total	4	7.088			
Response: 4th gelatin fraction yield; R^2^ = 99.20%					
Regression	3	1.242	0.414	41.4	0.114
Factor A (Temp. of 1st fraction)	1	0.09	0.09	9.0	0.205
Factor B (Temp. of 2nd fraction)	2	1.152	0.576	57.6	0.093
Error	1	0.01	0.01		
Total	4	1.252			

*—statistically significant factor.

**Table 4 gels-12-00045-t004:** Weighted results of the gelatin production cycle impacts assessment according to EF 3.1.

Impact Category	Quantity [Points]	Contribution [%]
Freshwater eutrophication	4.91 × 10^−3^	29.212
Fossils	3.15 × 10^−3^	18.725
Climate change	2.87 × 10^−3^	17.106
Minerals and metals	1.17 × 10^−3^	6.960
Ionizing radiation	9.24 × 10^−4^	5.497
Human toxicity, non-cancer	8.40 × 10^−4^	5.000
Freshwater and terrestrial acidification	6.99 × 10^−4^	4.158
Freshwater ecotoxicity	5.99 × 10^−4^	3.567
Water scarcity	4.67 × 10^−4^	2.781
Photochemical ozone creation	3.95 × 10^−4^	2.350
Marine eutrophication	2.43 × 10^−4^	1.445
Terrestrial eutrophication	2.01 × 10^−4^	1.197
Respiratory effects, inorganics	1.61 × 10^−4^	0.955
Human toxicity, cancer	1.42 × 10^−4^	0.846
Land use	3.22 × 10^−5^	0.192
Ozone layer depletion	1.41 × 10^−6^	0.008

**Table 5 gels-12-00045-t005:** Contributions of the gelatin individual production processes to three significant impact categories (weighted results).

Process	Freshwater Eutrophication	Fossils	Climate Change
Pigment extraction and gelatin product drying	3.33 × 10^−3^	2.12 × 10^−3^	1.94 × 10^−3^
Gelatin extraction	1.49 × 10^−3^	9.47 × 10^−4^	8.70 × 10^−4^
Conditioning	3.70 × 10^−5^	2.38 × 10^−5^	2.18 × 10^−5^
Demineralization	2.70 × 10^−5^	1.78 × 10^−5^	1.63 × 10^−5^
Washing and drying	1.01 × 10^−5^	6.67 × 10^−6^	6.18 × 10^−6^
Defatting	1.86 × 10^−6^	1.18 × 10^−6^	1.08 × 10^−6^
Milling	2.21 × 10^−7^	1.40 × 10^−7^	1.29 × 10^−7^
Hexane recycling	1.40 × 10^−6^	2.78 × 10^−5^	8.59 × 10^−6^
Vacuum evaporation	1.17 × 10^−5^	7.46 × 10^−6^	6.85 × 10^−6^

**Table 6 gels-12-00045-t006:** Weighted results of impact categories for the reference scenario and the scenario using a greener energy mix.

Impact Category	Reference Scenario [Points]	Greener Energy [Points]	Percentage Change [%]
Freshwater eutrophication	4.91 × 10^−3^	1.58 × 10^−3^	67.85
Fossils	3.15 × 10^−3^	2.20 × 10^−3^	30.20
Climate change	2.87 × 10^−3^	1.43 × 10^−3^	50.19
Minerals and metals	1.17 × 10^−3^	1.19 × 10^−3^	−1.48
Ionizing radiation	9.24 × 10^−4^	7.23 × 10^−4^	21.73
Human toxicity, non-cancer	8.40 × 10^−4^	6.17 × 10^−4^	26.59
Freshwater and terrestrial acidification	6.99 × 10^−4^	4.27 × 10^−4^	38.96
Freshwater ecotoxicity	5.99 × 10^−4^	2.97 × 10^−4^	50.43
Water scarcity	4.67 × 10^−4^	4.53 × 10^−4^	3.16
Photochemical ozone creation	3.95 × 10^−4^	2.57 × 10^−4^	34.93
Marine eutrophication	2.43 × 10^−4^	1.16 × 10^−4^	52.33
Terrestrial eutrophication	2.01 × 10^−4^	1.13 × 10^−4^	43.76
Respiratory effects, inorganics	1.61 × 10^−4^	1.59 × 10^−4^	0.89
Human toxicity, cancer	1.42 × 10^−4^	9.93 × 10^−5^	30.16
Land use	3.22 × 10^−5^	4.15 × 10^−5^	−28.85
Ozone layer depletion	1.41 × 10^−6^	1.68 × 10^−6^	−19.73

**Table 7 gels-12-00045-t007:** Weighted results of impact categories for the reference scenario and the scenarios of higher and lower hexane recycling ratios.

Impact Category	Ref. Scenario [Point]	82% Hexane rec. [Point]	Change [%]	42% Hexane Rec. [Point]	Change [%]
Freshwater eutrophication	4.91 × 10^−3^	4.91 × 10^−3^	0.01	4.91 × 10^−3^	−0.01
Fossils	3.15 × 10^−3^	3.13 × 10^−3^	0.47	3.16 × 10^−3^	−0.47
Climate change	2.87 × 10^−3^	2.87 × 10^−3^	0.16	2.88 × 10^−3^	−0.16
Minerals and metals	1.17 × 10^−3^	1.17 × 10^−3^	0.18	1.17 × 10^−3^	−0.18
Ionizing radiation	9.24 × 10^−4^	9.24 × 10^−4^	0.01	9.24 × 10^−4^	−0.01
Human toxicity, non-cancer	8.40 × 10^−4^	8.39 × 10^−4^	0.13	8.41 × 10^−4^	−0.13
Freshwater and terrestrial acidification	6.99 × 10^−4^	6.98 × 10^−4^	0.15	7.00 × 10^−4^	−0.15
Freshwater ecotoxicity	5.99 × 10^−4^	5.98 × 10^−4^	0.18	6.01 × 10^−4^	−0.18
Water scarcity	4.67 × 10^−4^	4.66 × 10^−4^	0.22	4.68 × 10^−4^	−0.22
Photochemical ozone creation	3.95 × 10^−4^	3.92 × 10^−4^	0.80	3.98 × 10^−4^	−0.80
Marine eutrophication	2.43 × 10^−4^	2.43 × 10^−4^	0.13	2.43 × 10^−4^	−0.13
Terrestrial eutrophication	2.01 × 10^−4^	2.01 × 10^−4^	0.18	2.01 × 10^−4^	−0.18
Respiratory effects, inorganics	1.61 × 10^−4^	1.60 × 10^−4^	0.62	1.61 × 10^−4^	−0.62
Human toxicity, cancer	1.42 × 10^−4^	1.42 × 10^−4^	0.15	1.42 × 10^−4^	−0.15
Land use	3.22 × 10^−5^	3.21 × 10^−5^	0.38	3.23 × 10^−5^	−0.38
Ozone layer depletion	1.41 × 10^−6^	1.39 × 10^−6^	1.20	1.42 × 10^−6^	−1.20

**Table 8 gels-12-00045-t008:** Weighted results of impact categories for the reference scenario and the scenarios of usage of demineralized water instead of tap water.

Impact Category	Ref. Scenario [Point]	Demineralized Water [Point]	Change [%]
Freshwater eutrophication	4.91 × 10^−3^	4.91 × 10^−3^	0.001
Fossils	3.15 × 10^−3^	3.15 × 10^−3^	0.001
Climate change	2.87 × 10^−3^	2.88 × 10^−3^	−0.009
Minerals and metals	1.17 × 10^−3^	1.17 × 10^−3^	−0.102
Ionizing radiation	9.24 × 10^−4^	9.24 × 10^−4^	0.014
Human toxicity, non-cancer	8.40 × 10^−4^	8.39 × 10^−4^	0.095
Freshwater and terrestrial acidification	6.99 × 10^−4^	6.99 × 10^−4^	−0.054
Freshwater ecotoxicity	5.99 × 10^−4^	6.08 × 10^−4^	−1.431
Water scarcity	4.67 × 10^−4^	4.68 × 10^−4^	−0.215
Photochemical ozone creation	3.95 × 10^−4^	3.95 × 10^−4^	−0.015
Marine eutrophication	2.43 × 10^−4^	2.43 × 10^−4^	−0.003
Terrestrial eutrophication	2.01 × 10^−4^	2.01 × 10^−4^	−0.005
Respiratory effects, inorganics	1.61 × 10^−4^	1.61 × 10^−4^	−0.142
Human toxicity, cancer	1.42 × 10^−4^	1.43 × 10^−4^	−0.222
Land use	3.22 × 10^−5^	3.22 × 10^−5^	−0.028
Ozone layer depletion	1.41 × 10^−6^	1.43 × 10^−6^	−1.799

**Table 9 gels-12-00045-t009:** The gel strength and dynamic viscosity of the second, third, and fourth gelatin fractions under different experimental conditions.

Experiment	Gel Strength [Bloom]			Dynamic Viscosity [mPa·s]		
	Fractions					
	2.	3.	4.	2.	3.	4.
1 (35/50 °C)	0	0	115.6 ± 10.1	1.05 ± 0.05	0.99 ± 0.02	2.24 ± 0.07
2 (35/60 °C)	168.8 ± 3.6	166.6 ± 12.9	167.8 ± 17.3	2.48 ± 0.02	3.44 ± 0.03	3.62 ± 0.11
3 (45/50 °C)	99.0 ± 2.6	118.2 ± 8.6	121.9 ± 11.9	1.71 ± 0.05	3.32 ± 0.02	3.23 ± 0.04
4 (45/60 °C)	79.4 ± 3.1	0	0	1.68 ± 0.01	0.76 ± 0.05	0.74 ± 0.02
5 (40/55 °C)	28.7 ± 1.2	1.6 ± 0.1	2.8 ± 0.3	1.65 ± 0.06	1.03 ± 0.03	1.05 ± 0.02
Control (40/55 °C)	0	0	73.7 ± 1.4	0.84 ± 0.08	1.04 ± 0.03	1.71 ± 0.05

**Table 10 gels-12-00045-t010:** The melting point and gelling point of the second, third, and fourth gelatin fractions under different experimental conditions.

Experiment	Melting Point [°C]			Gelling Point [°C]		
	Fraction					
	2.	3.	4.	2.	3.	4.
1 (35/50 °C)	0	0	25.0 ± 1.1	0	0	13.1 ± 0.7
2 (35/60 °C)	25.3 ± 0.3	28.3 ± 0.1	26.0 ± 0.2	13.1 ± 0.8	16.4 ± 0.6	14.3 ± 1.7
3 (45/50 °C)	23.2 ± 2.3	24.4 ± 0.5	25.6 ± 0.6	12.2 ± 0.1	12.2 ± 0.5	11.0 ± 0.6
4 (45/60 °C)	23.1 ± 2.2	0	0	12.9 ± 0.2	0	0
5 (40/55 °C)	22.3 ± 0.2	11.1 ± 0.2	11.3 ± 0.1	8.7 ± 1.1	5.5 ± 0.2	5.8 ± 0.3
Control (40/55 °C)	0	0	19.7 ± 3.5	0	0	10.6 ± 1.0

**Table 11 gels-12-00045-t011:** The fatty acid composition of ACBP oil.

Fatty Acid Name	Shorthand	Relative Percentage [%]
Myristic acid	C14:0	2.28 ± 0.02
Palmitic acid	C16:0	20.45 ± 0.09
Palmitoleic acid	C16:1n-7	4.19 ± 0.02
Stearic acid	C18:0	6.64 ± 0.05
Oleic acid	C18:1n-9	29.08 ± 0.12
Vaccenic acid	C18:1n-5	3.05 ± 0.03
Linoleic acid	C18:2n-6	14.60 ± 0.10
α-Linolenic acid	C18:3n-3	2.02 ± 0.06
Gadoleic acid	C20:1n-9	2.60 ± 0.02
Arachidonic acid	C20:4n-6	0.67 ± 0.03
Eicosapentaenoic acid	C20:5n-3	2.23 ± 0.06
Docosapentaenoic acid	C22:5n-3	0.89 ± 0.03
Docosahexaenoic acid	C22:6n-3	4.65 ± 0.07
ΣSFA		30.85 ± 0.04
ΣMUFA		40.52 ± 0.07
ΣPUFA		28.65 ± 0.06

## Data Availability

The original contributions presented in the study are included in the article; further inquiries can be directed to the corresponding author.

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
