# Peer review of "Gels2026, 12(1), 45;https://doi.org/10.3390/gels12010045"

_gels, 2026, doi:10.3390/gels12010045_

Round 1

Reviewer 1 Report

Comments and Suggestions for Authors

  1. Abstract lacks a clear hypothesis or research question. Please improve it
  2. The innovation claim is frail without conflicting with current valorization methods.
  3. The assembly between the biorefinery and aquaculture sustainability is weak. The industrial scalability of the process is not designated.
  4. Rationale for enzyme conditioning needs stronger mechanistic explanation.
  5. No evidence is provided on enzyme specificity or compatibility with AC collagen. Please devlop more this part
  6. Extraction temperature ranges may not capture full collagen thermal behavior.
  7. Zero-waste framework is theoretically strong but experimentally incomplete.
  8. LCA at laboratory scale may not reproduce industrial feasibility. Please more explain this part in the whole of text
  9. Environmental hotspots recognized are generic (water/energy) rather than AC-specific.
  10. Economic viability of multiproduct extraction pipeline is not addressed. Please more explain since authors have a patent
  11. Pigment extract recovery pathway lacks justification of functionality or value.
  12. Microbial enzyme selection criteria are not detailed.
  13. No explanation given for selecting specific temperature pairings (35/60 °C). more explain if all studied parameters levels
  14. Study does not quantify variability in AC by-product composition.
  15. Limited discussion on scalability of the sequential extraction steps.
  16. No risk assessment for microbial contamination during enzymatic conditioning. More explain this point
  17. Claim of circular bioeconomy alignment lacks measurable indicators.
  18. More address regulatory implications for multi-stream bioproducts.
  19. General point: some techniques in MM section should be concise
  20. Some figs should be inserted as a sup. Files

Reviewer 2 Report

Comments and Suggestions for Authors

The authors deal with an interesting subject like the valorization of African Catfish by-products, paying special attention to LCA, but also providing an extensive characterization of different products obtained. In general, I consider this work innovative, and the approach given by the authors can be interesting to this journal. 

Having said that, there are some suggestions that could enhance the quality of this article, like the following:

  • The authors can remove "African Catfish" in the keywords section, as it is already included in the title. You can replace this keyword for another one to increase the impact of your article in search engines.
  • Introduction: It is OK, especially focused on this issue and providing interesting data that support the purpose of this work. I only recommend to move the reasoning about LCA after the explanation of the different products (gelatin, oil, pigment...), and before the aim of the work. It sounds more logic in my opinion.
  • Materials and Methods: The methodology is well explained, giving a lot of details. I only have one suggestion, about ACBPs characterization. First, a Table with the main characteristics of compositional analysis (and the corresponding references) would be interesting instead of giving a long paragraph. Also, the results should be given in the "Results and Discussion section", in my opinion. The quality of Figure 6 could be improved, as it can not be seen properly (at least in the version I have received). Finally, the details of gas chromatography could be better presented in a table, with special focus on carrier gas and temperature program.
  • Results and Discussion. This is a very interesting part, which is highly described and reasoned by the authors. Concerning LCA, interesting findings are shown, which can not be supported by the literature due to the specific study carried out. This is one of the main problems in LCA studies, that is, they can not be properly compared to anything else due to their specific nature. Nevertheless, I would encourage the authors to try and find references where the main foundations of LCA are exposed, providing at least a general support to the findings obtained in this work. For instance, water consumption and energy demand are the most representative environmental challenges in many different biorefineries, and providing some examples (although not highly related to your proposal) could be an interesting point. I have to congratulate the authors about the sections where practical implications, limitations and future prospects are given. They really enhance the quality and rigor of this work.

Round 2

Reviewer 1 Report

Comments and Suggestions for Authors

accept